# META-LEARNING INITIALIZATIONS FOR IMAGE SEGMENTATION

## ABSTRACT

While meta-learning approaches that utilize neural network representations have made progress in few-shot image classification, reinforcement learning, and, more recently, image semantic segmentation, the training algorithms and model architectures have become increasingly specialized to the few-shot domain. A natural question that arises is how to develop learning systems that scale from few-shot to many-shot settings while yielding competitive performance in both. One scalable potential approach that does not require ensembling many models nor the computational costs of relation networks, is to meta-learn an initialization. In this work, we study first-order meta-learning of initializations for deep neural networks that must produce dense, structured predictions given an arbitrary amount of training data for a new task. Our primary contributions include (1), an extension and experimental analysis of first-order model agnostic meta-learning algorithms (including FOMAML and Reptile) to image segmentation, (2) a novel neural network architecture built for parameter efficiency and fast learning which we call EfficientLab, (3) a formalization of the generalization error of meta-learning algorithms, which we leverage to decrease error on unseen tasks, and (4) a small benchmark dataset, FP-k, for the empirical study of how meta-learning systems perform in both few- and many-shot settings. We show that meta-learned initializations for image segmentation provide value for both canonical few-shot learning problems and larger datasets, outperforming random and ImageNet-trained initializations for up to 400 densely labeled examples. Finally, we show both theoretically and empirically that a key limitation of MAML-type algorithms is that when adapting to new tasks, a single update routine is used that is not conditioned on the available data for a new task. We find that our network, with an empirically estimated optimal update procedure yields state of the art results on the FSS-1000 dataset, while only requiring one forward pass through a single model at evaluation time.

## 1 INTRODUCTION

Humans have a remarkable capability to not only learn new concepts from a small number of labeled examples but also to gain expertise as more data becomes available. In recent years, there has been substantial progress in high accuracy image segmentation in the high data regime (see Liu et al. (2019) and their references). While meta-learning approaches that utilize neural network representations have made progress in few-shot image classification, reinforcement learning, and, more recently, image semantic segmentation, the training algorithms and model architectures have become increasingly specialized to the low data regime. A desirable property of a learning system is one that effectively applies knowledge gained from a few *or* many examples, while reducing the generalization gap when trained on little data and not being encumbered by its own learning routines when there are many examples. This property is desirable because training and maintaining multiple models is more cumbersome than training and maintaining one model. A natural question that arises is how to develop learning systems that scale from few-shot to many-shot settings while yielding competitive accuracy in both. One scalable potential approach that does not require ensembling many models nor the computational costs of relation networks, is to meta-learn an initialization.

In this work, we specifically address the problem of meta-learning initializations for deep neural networks that must produce dense, structured output, such as for the semantic segmentation of images. We ask the following questions:

1. Do first-order MAML-type algorithms extend to the higher dimensional parameter spaces, dense prediction, and skewed distributions required of semantic segmentation?

2. How robust are the representations that the model has meta-learned to perturbations in the hyperparameters of the update routine?

3. Are MAML-type algorithms hindered by having a fixed update policy for training and testing tasks that is not conditioned on the available labeled examples for a new task?

4. What is the number of labeled examples beyond which a conventional approach to training deep neural networks outperforms the meta-learned initializations?

To address the fourth question, we put together a small benchmark dataset, which we call FP-k, that contains 400 training examples for 5 tasks each. In recent works (Li et al., 2017; Shaban et al., 2017; Rusu et al., 2018; Zhang et al., 2019; Lee et al., 2019), few-shot learning approaches have become increasingly complex and appear to be specialized to the few-shot domain. This raises many open questions, such as: What is the accuracy of a few-shot learning system when more labeled examples become available? After a certain number of labeled examples, will the few-shot learning system have the same accuracy as a simpler training approach such as conventional training via SGD? If so, what is the number of labeled examples beyond which a conventional approach to training deep neural networks outperforms a meta-learning system? We address these questions in 5.4, providing empirical justification for our meta-learning approach for up to 400 densely labeled examples.

Through a series of theoretical and empirical analyses, we shed new light on the representations that model agnostic meta-learning algorithms learn and how they adapt to unseen tasks. In summary, we address the above research questions as follows: We show that MAML-type algorithms do extend to few shot image segmentation, yielding state of the art results when their update routine is optimized after meta-training and when the model is regularized. Addressing question 2, we find that the meta-learned initialization's performance when being evaluated on a task is particularly sensitive to changes in the update routine's hyperparameters (see Figure 2). We show theoretically in 3.2 and empirically in our results (see Table 2) that a fixed update routine does hinder generalization performance. Finally, we address question 4 by showing that our meta-learned initializations are competitive with ImageNet (Deng et al., 2009) trained initializations for up to 400 labeled examples. We will release our code and the FP-k dataset upon acceptance.

## 2 RELATED WORK

Learning useful models from a small number of labeled examples of a new concept has been studied for decades (Thrun, 1996) yet remains a challenging problem with no semblance of a unified solution. The advent of larger labeled datasets containing examples from many distinct concepts (Vinyals et al., 2016) has enabled progress in the field in particular by enabling approaches that leverage the representations of nonlinear neural networks. Image segmentation is a well-suited domain for advances in few-shot learning given that the labels are particularly costly to generate (Wei et al., 2019).

Recent work in few-shot learning for image segmentation has utilized three key components: (1) model ensembling (Shaban et al., 2017), (2) relation networks (Santoro et al., 2017), and (3) late fusion of representations (Rakelly et al., 2018; Wei et al., 2019). The inference procedure of ensembling models with a separately trained model for each example has been shown to produce better predictions than single shot approaches but will scale linearly in time and/or space complexity (depending on the implementation) in the number of training examples, as implemented in Shaban et al. (2017). An exciting recent approach is that of using relation networks to learn modules that can reason about the relationships of examples in a domain (Santoro et al., 2017). Relation networks have seen increased adoption in meta-learning systems (Rusu et al., 2018) and were recently employed in few-shot segmentation in Zhang et al. (2019) and Wei et al. (2019). While relation networks yield impressive results in the few-shot domain, they typically come with $O(n^2)$ training costs where $n$ is the number of support and query examples. The use of multiple passes through subnetworks via iterative optimization modules used by Zhang et al. (2019) further exacerbate these costs. The authors in Wei et al. (2019) take an elegant approach to reducing the computational complexity by element-wise summing all feature maps of the support examples over the channel dimension, such that the encoded feature map tensors have the same depth regardless of the size of the support set, forming a type of embedded memory of the class and then relating that to the query examples. While

this approach reduces computational complexity of traditional applications of relation networks, its efficacy in scaling as more training data becomes available remains unclear.

Model Agnositc Meta-Learning (MAML) is a gradient-based meta-learning approach introduced in Finn et al. (2017). First Order MAML (FOMAML) reduces the computational cost by not requiring backpropogating the meta-gradient through the inner-loop gradient and has been shown to work similarly well on classification tasks (Finn et al., 2017; Nichol & Schulman, 2018). Though learning an initialization has the potential to unify few-shot and many-shot domains, initializations learned from MAML-type algorithms have been seen to overfit in the low-shot domain when adapting sufficiently expressive models such as deep residual networks that may be more than a small number of convolutional layers [1] (Mishra et al., 2018; Rusu et al., 2018). The Meta-SGD learning framework added additional capacity to the same network architecture used in MAML with improved generalization by meta-learning a learning rate for each parameter in the network (Li et al., 2017), but lacks a first order approximation. In addition to possessing potential to unify few- and many-shot domains, MAML-type algorithms are intriguing in that they impose no constraints on model architecture, given that the output of the meta-learning process is simply an initialization. Futhermore, the meta-learning dynamics, which learn a temporary memory of a sampled task, are related to the older idea of fast weights (Hinton & Plaut, 1987; Ba et al., 2016). Despite being dataset size and model architecture agnostic, MAML-type algorithms are unproven for high dimensionality of the hypothesis spaces and the skewed distributions of image segmentation problems data (Rakelly et al., 2018). In this work, we show, for the first time, that model agnostic meta-learning algorithms do in fact naturally extend to image segmentation, yielding state of the art results when their update procedure is optimized.

## 3 PRELIMINARIES

We assume access to a distribution over tasks $\hat{p}(\mathcal{T})$ that is sampled from a generating distribution $p(\mathcal{T})$. In the context of image segmentation, an example from a task $\mathcal{T}_i$ is comprised of an image $x$ and its corresponding binary masks $y$, which assign each pixel mutually exclusive membership to the target (for example, black bear) or background class. Following the notation put forth in Rusu et al. (2018), the tasks in $\hat{p}(\mathcal{T})$ are divided into disjoint meta-training $\mathcal{S}^{tr}$, validation $\mathcal{S}^{val}$, and test $\mathcal{S}^{test}$ sets. The objective is to find a set of optimal model parameters $\theta^*$ that minimize the loss $\mathcal{L}$ of predictions $\hat{y}$ with respect to the distribution of examples $q_{\mathcal{T}_i}(x, y)$ for a new task $\mathcal{T}_i$. To estimate $\theta^*$ in practice, we apply an update procedure $U$ with hyperparameters $\omega$ to train a model on the labeled examples $\mathcal{D} := \left\{ x^{(k)}, y^{(k)} \right\}$ from $q_{\mathcal{T}_i}(x, y)$.

### 3.1 MODEL AGNOSTIC META-LEARNING

The meta-learning algorithms introduced in Finn et al. (2017) and Nichol & Schulman (2018), work by sampling a few-shot dataset $\mathcal{D}_s$ from a task $\mathcal{T}_i$, adapting a model's internal representations, $\theta$, for a small number of gradient updates, $j$, to produce $\theta_i'$ after $j$ steps of gradient descent. We define this update routine as the function $U$. The objective is to find an optimal *initialization* $\theta_0^*$ which minimizes loss after training on a new task:

$$\theta_0^* = \arg\min_{\theta_0} \mathbb{E}_p \left[ \mathbb{E}_{q_{\mathcal{T}_i}} \left[ \mathcal{L} \left( U(\theta_0 ; \mathcal{D}_s, \omega) \right) \right] \right] \tag{1}$$

Taking the derivative with respect to $\theta_0$

$$\frac{\partial}{\partial \theta_0} \mathcal{L}(U(\theta_0)) = U'(\theta_0) \cdot \mathcal{L}'(U(\theta_0)) \tag{2}$$

results in the term $U'$, which is the derivative of a gradient-based update procedure and, hence, contains second order derivatives.

In first-order renditions, finite difference is used to approximate the gradient of the meta-update between $\theta_i'$ and $\theta_0$. FOMAML differs from Reptile in that $\mathcal{D}_s$ is split into mini-training, $\mathcal{D}_{tr}$, and

---

[1]The original MAML and Reptile convolutional neural networks (CNNs) use four convolutional layers with 32 filters each for MiniImagenet (Finn et al., 2017; Nichol & Schulman, 2018)

validation, $\mathcal{D}_{val}$, sets. The last batch of the inner-loop is used to "correct" for overfitting to $\mathcal{D}_{tr}$ by taking a gradient step in the direction of descent pointed to by the loss, $\mathcal{L}$, on $\mathcal{D}_{val}$. The difference between the two approximations can be summarized by how they make use of two update steps – an initial update using the training examples, and a second using the validation examples:

$$\theta \leftarrow U(\theta_0 \, ; \mathcal{D}_{tr} \, , \omega_{tr}) \tag{3}$$

$$\theta' \leftarrow U(\theta \, ; \mathcal{D}_{val}, \omega_{val}) \tag{4}$$

Reptile only makes use of the initial update in its first order difference, where FOMAML uses both

$$\text{Reptile: } \nabla\theta_0 \propto \theta - \theta_0 \tag{5}$$

$$\text{FOMAML: } \nabla\theta_0 \propto \theta' - \theta \tag{6}$$

This gradient approximation can then be used to optimize the initialization via stochastic gradient descent:

$$\theta_0 \leftarrow \theta_0 + \epsilon \cdot \nabla\theta_0 \tag{7}$$

or any other gradient-based update procedure.

## 3.2 GENERALIZATION ERROR OF META-LEARNING ALGORITHMS

The generalization gap between the error on the generating distribution and the empirical data in meta-learning is twofold: from the domain of all tasks $\mathcal{T}$ to the sample $\mathcal{T}_{tr}$, and within that, from all examples in $\mathcal{T}_i$ to $\mathcal{D}_s$. Taking $\hat{p}$ as $p(\mathcal{T}_i | \mathcal{T}_i \in \mathcal{T}_{tr})$, and similarly, $\hat{q}_{\mathcal{T}_i}$ as $q_{\mathcal{T}_i}(x, y)|(x, y) \in \mathcal{D}_s)$, we approximate the empirically optimal initialization

$$\hat{\theta}_0^* = \arg\min_{\theta_0} \mathbb{E}_{\hat{p}} \left[ E_{\hat{q}_{\mathcal{T}_i}} \left[ \mathcal{L} \left( U(\theta_0 \, ; \mathcal{D}_s, \omega) \right) \right] \right] \tag{8}$$

The generalization error can then be expressed as

$$\mathbb{E}_p \left[ \mathbb{E}_{q_{\mathcal{T}_i}} \left[ \mathcal{L} \left( U(\hat{\theta}_0^*) \right) \right] \right] - \mathbb{E}_{\hat{p}} \left[ \mathbb{E}_{\hat{q}_{\mathcal{T}_i}} \left[ \mathcal{L} \left( U(\hat{\theta}_0^*) \right) \right] \right] \tag{9}$$

To reduce the generalization gap in practice, model agnostic meta-learning approaches must address both generalization across tasks, and within tasks from the limited examples available. Because the loss in 1 is defined in terms of the update routine $U$ and given that $U$ in practice is parameterized by both a sampled dataset $D_s$ *and* $\omega$, there will also be an optimal set of hyperparameters, $\omega^*$ for every $\theta_0$, which may or may not be equal to $\omega$. Thus, the expectation of the loss with respect to $\omega$ given a fixed $\theta_0$ can also be minimized:

$$\hat{\omega}^* = \arg\min_{\omega} \mathbb{E}_{\hat{p}} \left[ \mathbb{E}_{\hat{q}_{\mathcal{T}_i}} \left[ \mathcal{L} \left( U(\omega \, ; \theta_0, \mathcal{D}_s) \right) \right] \right] \tag{10}$$

Thus, on many problems, it may be useful to estimate an optimal update procedure on the available data given the meta-learned initialization. We apply this insight to developing a simple update hyperparameter optimization (UHO) algorithm which we find significantly improves performance on the test set of tasks $S^{test}$. We discuss hyperparameter tuning further in 5.3. The algorithm is shown in full pseudocode in the appendix in D.

## 4 EFFICIENTLAB ARCHITECTURE FOR IMAGE SEGMENTATION

To extend first-order MAML-type algorithms to more expressive models, with larger hypothesis spaces, while yielding state of the art few-shot learning results, we developed a novel neural network architecture, which we term EfficientLab. The top level hierarchy of the network's organization of computational layers is similar to Chen et al. (2018), with 4 convolutional blocks that successively halve the features in spatial resolution while increasing the number of feature maps. This is followed by a 4x bilinear upsampling which is concatenated with features from a long skip connection from the second downsampling block in the encoding part of the network. The concatenated low and high resolution features are then fed through an atrous spatial pyramid pooling (ASPP) module and finally bilinearly upsampled to original image size.

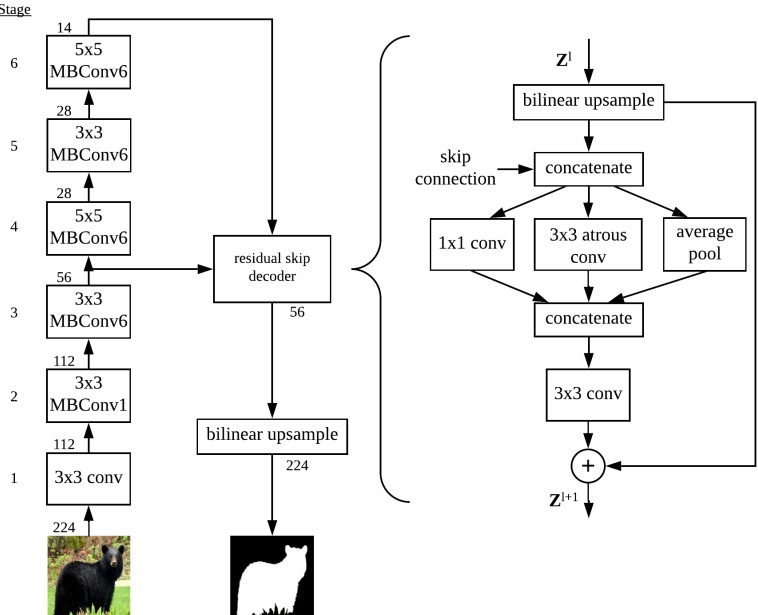

Figure 1: Diagram of the computations performed by the EfficientLab neural network. Nodes represent functions and edges represent output tensors. Output spatial resolutions are written next to the output edge. The high level architecture shows the EfficientNet feature extractor on the left with mobile inverted bottleneck convolutional blocks (see Tan & Le (2019); Sandler et al. (2018) for more details). On the right is the residual skip decoder module that we utilize in the upsampling branch of EfficientLab.

The differences between our model and the DeepLab model are in (1) the encoder network used and (2) how the low resolution embedded features are upsampled to full resolution predictions. For the encoding subnetwork, we utilize the recently proposed EfficientNet (Tan & Le, 2019). After encoding the images, instead of feeding them directly into an atrous spatial pyramid pooling module (ASPP), we first immediately bilinearly upsample the features by 4x. The upsampled features are then concatenated with features from the second downsampling block. Moving the ASPP module to the usampled resolution provides two advantages. First, it allows us to use 1 convolutional module in place of two. Due to the high dimensionality of the features along the channel axis at the lower resolution feature maps, the convolutional kernels are especially expensive in terms of number of parameters. Second, the ASPP module is designed to learn multiscale context which could be useful in refining the boundaries of semantic features in mid-resolution feature maps. Our ASPP module utilizes three parallel branches of a $1 \times 1$ convolution, $3 \times 3$ convolution with dilation rate = 2, and a simple average-pooling across spatial dimensions of the feature maps. The output of the three branches is concatenated and fed into a final $3 \times 3$ convolutional layer with 112 filters. A residual connection wraps around the convolutional layers to ease gradient flow [2]. We call this structure a residual skip decoder (RSD) and its computational graph of operations is shown in Figure 1. Before the final $1 \times 1$ convolution that produces the unnormalized heatmap of class scores, we use a single layer of dropout with a drop rate probability = 0.2 [3]. We use the standard softmax to produce the normalized predicted probabilities.

We use batch normalization layers following convolutional layers (Ioffe & Szegedy, 2015). We meta-learn the $\beta$ and $\gamma$ parameters, adapt them at test time to test tasks, and use running averages as estimates for the population mean and variance, $E[x]$ and $Var[x]$, at inference time as suggested in

---

[2]Residual connections have been suggested to make the loss landscape of deep neural networks more convex (Li et al., 2018). If this is the case, it could be especially helpful in finding low-error minima via gradient-based update routines such as those used by MAML, FOMAML, and Reptile.

[3]As described in Li et al. (2019) and used in Tan & Le (2019) the dropout layer is applied after all batch norm layers.

Antoniou et al. (2018). *All* parameters at the end of an evaluation call are reset to their pre-adaptation values to stop information leakage between the training and validation sets. The network is trained with the binary cross entropy minus the log of the dice score (Dice, 1945) , which we adapt from the loss function of (Iglovikov et al., 2017), plus an $L_2$ regularization on the weights:

$$\mathcal{L} = H - log(\mathcal{J}) + \lambda \left\| \theta \right\|_2^2 \tag{11}$$

where $H$ is binary cross entropy loss:

$$H = -\frac{1}{n} \sum_{i=1}^{n} \left( y_i \log \hat{y}_i + (1 - y_i) \log (1 - \hat{y}_i) \right) \tag{12}$$

$J$ is the modified Dice score:

$$J = \frac{2IoU}{IoU + 1} \tag{13}$$

and $IoU$ is the intersection over union metric:

$$IoU = \frac{1}{n} \sum_{i=1}^{n} \left( \frac{y_i \hat{y}_i + \epsilon}{y_i + \hat{y}_i - y_i \hat{y}_i + \epsilon} \right) \tag{14}$$

## 5 EXPERIMENTS

We evaluate the FOMAML and Reptile meta-learning algorithms on the FSS-1000 dataset. Model topology development[4] and hyperparameter search was done on a held out set of validation tasks $S^{val}$ and not the final test tasks. For the final evaluations, we meta-train for $\sim$200 epochs through the training and validation tasks, $S^{tr} \cup S^{val}$, using a meta-batch size of 5, an inner loop batch size of 8, and 5 inner loop iterations. During training, we use stochastic gradient descent (SGD) in the inner loop with a fixed learning rate of 0.005. During training and evaluation, we apply simple augmentations to the few-shot examples including random translation, horizontal flips, additive Gaussian noise, brightness, and random eraser (Zhong et al., 2017). We use $L_2$ regularization on all weights with a coefficient $\lambda = 5\mathrm{e}{-4}$.

### 5.1 DATASETS

The first few-shot image segmentation dataset was the PASCAL-5[i] presented in Shaban et al. (2017) which reimagines the PASCAL dataset (Everingham et al., 2010) as a few-shot binary segmentation problem for each of the classes in the original dataset. Unfortunately, the dataset contains relatively few distinct tasks (20 excluding background and unlabeled). The idea of a meta-learning dataset for image segmentation was further developed with the recently introduced FSS-1000 dataset, which contains 1000 classes, 240 of which are dedicated to the meta-test set $S^{test}$, with 10 image-mask pairs for each class (Wei et al., 2019). For each of the rows in the results table 2, we evaluate the network on the 240 test tasks, sampling two random splits into training and testing sets for each task, yielding 480 data points per meta-learning approach for which the mean intersection over union (IoU) (eq. 14) and 95% confidence interval are reported. The FSS-1000 dataset is the focus of the empirical comparisons of network ablations and meta-learning approaches that we experiment with in this paper.

### 5.2 FP-K DATASET

For investigating how the meta-learned representations integrate new information as more data becomes a available, we put together a small benchmark dataset that we call FP-k. FP-k takes 5 tasks from FSS-1000 and 5 tasks from PASCAL-5[i] for the same concept[5]. Using this dataset, we train over a range of "k"-training shots from ImageNet-trained initializations[6] and our meta-learned initializations. We report the performance of our EfficientLab network meta-trained with FOMAML

---

[4]Though extensive model experimentation was done on $S^{val}$, we report ablations on the meta-test set $S^{test}$ in the results section for comparison to our best methods.

[5]See the Appendix for more details on the dataset construction.

[6]The encoder is trained on ImageNet, while the residual skip decoder and final layer weights are initialized using the Glorot uniform initialization (Glorot & Bengio, 2010)

over a range of k examples as a benchmark which we hope will inspire future empirical research into studying how meta-learning approaches scale in accuracy and computational complexity as more labeled data become available. These results are show in Figure 3 and discussed in 5.4.

## 5.3 HYPERPARAMETER SELECTION

Generalization in meta-learning requires both the ability to learn representations for new tasks efficiently ($\mathcal{T}_{tr}$ to $\mathcal{T}_{test}$) from few training examples, and to select representations that are able to capture unseen test examples effectively ($\mathcal{D}_{tr}$ to $\mathcal{D}_{test}$). The approximation scheme of FOMAML addresses the latter by taking the finite difference between updates using the train and validation sets in 6, favoring initializations that differ less between splits of $\mathcal{D}_s$. To address sample efficiency for learning new tasks, we leverage the flexibility to choose hyperparameters $\omega_{tr}$ and $\omega_{test}$ separately.

Empirical estimations of the optimal initialization have an implicit dependence on $\mathcal{T}_{tr}$ and $\omega$ (eq. 8), and the optimal hyperparameters $\omega^*$ depend on the initialization $\theta_0$ in turn (eq. 10). For this reason, after learning $\hat{\theta}_0^*$, we then fix $\hat{\theta}_0^*$ and tune $\omega_{train}$ to find $\omega_{test}$. Selecting a small learning rate in training $\alpha_{tr}$ leads to initializations $\hat{\theta}_0^*$ close to parameters with low loss $\hat{\theta}_{\mathcal{T}_i}^*$ for tasks in $\mathcal{T}_{tr}$ (see Nichol & Schulman (2018) for further discussion). But, because the hyperparameters $\omega_{tr}$ that govern the dynamics of the update routine that is used to adapt to new tasks are not learned nor conditioned on $\mathcal{D}_s$, $\omega_{tr}$ is not guaranteed to minimize the empirical risk even over the training tasks in $S^{tr}$.

In our experiments, after we have meta-learned an initialization $\theta_0$, we then applied the UHO algorithm to tune the update routine's hyperparameters on 100 randomly sampled tasks from the training set. We specifically search over the learning rate and the number of gradient updates that are applied when adapting to a new task $\mathcal{T}_i$.

## 5.4 RESULTS

We show the results of experimenting with different decoder architectures for EfficientLab in Table 1 Each network topology is meta-trained with FOMAML and the same meta-training hyperparameters defined in 5.

| Network Architecture | $\overline{\text{IoU}}$ |
|---|---|
| EfficientNet w/o decoder | $75.66 \pm 1.01\%$ |
| EfficientNet + Auto-DeepLab decoder | $73.054 \pm 01.09\%$ |
| EfficientNet + RSD at Stage 3 w/o residual | $78.17 \pm 1.02\%$ |
| EfficientNet + RSD at Stages 3 & 6 | $80.11 \pm 0.94\%$ |
| EfficientNet + RSD at Stage 3 | $\mathbf{80.60 \pm 0.93\%}$ |

Table 1: EfficientLab architecture ablations. Each network is meta-trained in the same way following 5 and tested on the set of test tasks from FSS-1000 (Wei et al., 2019). The 3rd row contains results of removing the short-range residual connection from our proposed RSD module. The final row is the best network we find for few-shot performance via model agnostic meta-learning. We call this network EfficientLab in reference to the encoder of EfficientNet (Tan & Le, 2019) and the decoder of Auto-DeepLab (Chen et al., 2018), which it is inspired by.

The results of our model with an initialization meta-learned using Reptile and FOMAML are shown in Table 2. We find that our model trained with FOMAML and importantly with regularization and improved use of batch normalization yields state of the art results. Given that previous works have used regularization minimally or not at all during meta-training, we also conducted an ablation of removing regularization on the model. We find, unsurprisingly, that the combination of an $L2$ loss on the weights, with simple augmentations, and a final layer of dropout does significantly increase generalization performance. After optimizing the update hyperparameters, our approach sets the new state of the art for the FSS-1000 dataset. We have included a visualization of example predictions for a small set of randomly sampled test tasks in A.

To address question 3 in section 1, we also searched through a range of update routine learning rates, $\alpha$, that were $10\times$ less to $10\times$ greater than the learning rate used during meta-training. As

| Method | $\overline{\text{IoU}}$ |
|---|---|
| FSS-1000 Baseline | 73.47% |
| **FOMAML** | $\mathbf{75.19 \pm 01.28\%}$ |

(a) FSS-1000 1-shot

| Method | $\overline{\text{IoU}}$ |
|---|---|
| FSS-1000 Baseline | 80.12% |
| Reptile | $62.36 \pm 2.12\%$ |
| FOMAML | $77.89 \pm 1.03\%$ |
| FOMAML + regularization | $80.60 \pm 0.93\%$ |
| **FOMAML + regularization + UHO** | $\mathbf{82.19 \pm 0.91\%}$ |

(b) FSS-1000 5-shot

Table 2: Mean IoU scores of the EfficientLab network evaluated on FSS-1000 test set of tasks for 1-shot and 5-shot learning. We report the FSS-1000 baseline from (Wei et al., 2019). Our best found model combined FOMAML, EfficientLab, regularization, and the UHO algorithm.

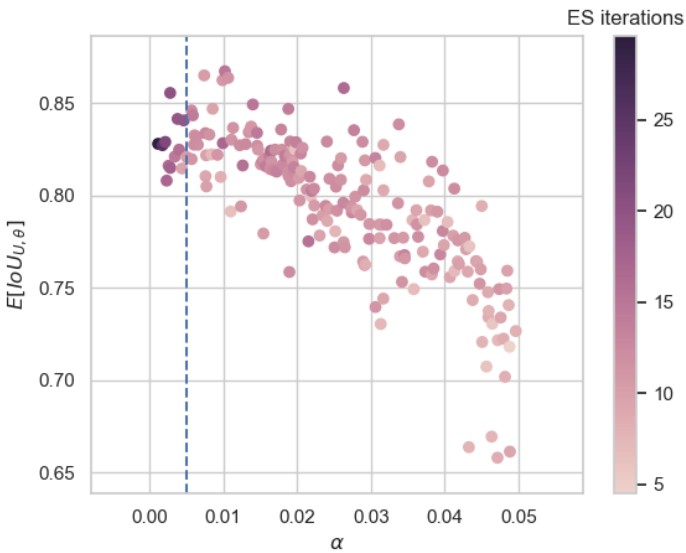

Figure 2: Each point represents the mean IoU for the validation tasks with a sampled learning rate $\alpha$. The blue dashed line indicates the learning rate used by SGD in the inner loop during meta-training. Points are colored by how many iterations they were trained before stopped by early stopping (ES) with a patience of 3 iterations.

clearly shown in Figure 2, the learned representations are **not** robust to such large variations in the hyperparameter.

In this work we posit that a fixed update procedure that is used at test time and not conditioned on the labeled examples for a new task $\mathcal{D}_{tr} \in \mathcal{T}_i$ is one of the major hinderances of apply MAML-type algorithms to unseen tasks. In section 3.2, we show that MAML-type algorithms do not natively guarantee that the update routine's hyperparameters are optimal. We find this analysis to be supported empirically as well. We find that: (1) the estimated optimal hyperparameters for the update routine even on the *training* tasks are not the same as those specified a priori during meta-training, as illustrated in Figure 2. One may expect that MAML-type algorithms would converge to a point in parameter space from which optimal minima for each of the training tasks are reachable. We find that even after 200 epochs through the training set, this was not the case. The best learning rate and number of iterations we found via the random search UHO algorithm were 0.007475 and 8, respectively, compared to a learning rate of 0.005 and 5 iterations used during training. (2) Optimizing the hyperparameters (even on the set of training tasks $S^{tr}$) after meta-training improves test-time results on unseen tasks. Furthermore, we find that *meta-training* from scratch and evaluating with the UHO-selected hyperparameters learning rate = 0.007475 and inner-iterations = 8 yields nearly identical results to meta-training with the initial hyper parameters learning rate = 0.005 and inner-

| Initialization | k | $\overline{\text{IoU}}$ |
|---|---|---|
| ImageNet | 1 | $16.76 \pm 4.62\%$ |
| ImageNet | 5 | $24.50 \pm 5.09\%$ |
| ImageNet | 10 | $27.08 \pm 5.58\%$ |
| ImageNet | 50 | $31.37 \pm 6.43\%$ |
| ImageNet | 100 | $37.39 \pm 6.66\%$ |
| ImageNet | 200 | $47.50 \pm 6.82\%$ |
| ImageNet | 400 | $55.86 \pm 7.15\%$ |
| Meta-learned | 1 | $38.30 \pm 9.42\%$ |
| Meta-learned | 5 | $42.59 \pm 9.16\%$ |
| Meta-learned | 10 | $43.94 \pm 9.86\%$ |
| Meta-learned | 50 | $50.12 \pm 8.62\%$ |
| Meta-learned | 100 | $53.37 \pm 7.90\%$ |
| Meta-learned | 200 | $55.55 \pm 7.73\%$ |
| Meta-learned | 400 | $58.68 \pm 7.48\%$ |

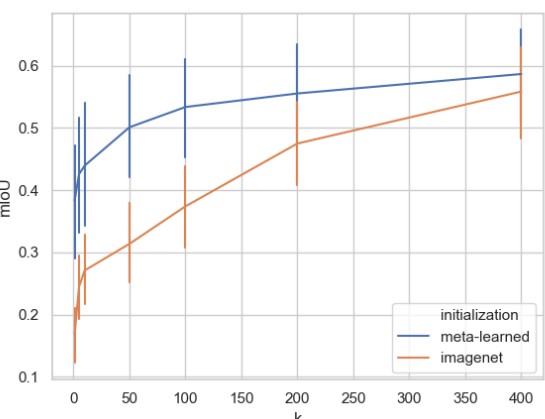

Figure 3: Mean IoU results as a function of the training set size of our EfficientLab model adapted to tasks of the FP-k dataset. Error bars represent 95% confidence intervals. The meta-learned initialization utilizes the top performing learning algorithm in Table 2. The meta-learned initialization outperformed EfficientLab initialized with an ImageNet-trained encoder and a randomly initialized decoder for all numbers of labeled training examples that we evaluated. For additional experimental details see C in the appendix.

iterations = 5. This further suggests that it may be useful to tune the hyperparameters $\omega$ to improve the generalization performance of the gradient-based adaptation routine $U$.

By training our model on the FP-k dataset, we also found that our meta-learned initializations outperformed an ImageNet-trained encoder and a randomly initialized decoder for up to 400 training examples[7].

## 6 DISCUSSION

In this work, we showed that gradient-based first order model agnostic meta-learning algorithms do in fact extend to the high dimensionality of the hypothesis spaces and the skewed distributions of few-shot image segmentation problems, but are sensitive to the hyperparameters, $\omega$, of the update routine, $U$. Furthermore, we find that the representations that are meta-learned are valuable as more data becomes available, unifying few- and many-shot regimes. We have also presented a novel neural network architecture, EfficientLab, for semantic image segmentation.

Future work should investigate more critically, both empirically and theoretically, the efficacy of few-shot learning systems as more labeled data becomes available. To this end, we have reported baseline results on a small meta-test benchmark dataset, FP-k, which contains 5 tasks with 400 training and 20 test examples per task. Additionally, future work could investigate more deeply learned update procedures, forms of meta-regularization, and second order methods for image segmentation. It would also be useful in future work to take a more critical look at the interplay between batch normalization and meta-learning. While single task deep neural networks in large data regimes apply batch normalization with a consistent pattern, different groups working in few-shot meta-learning have incorporated batch norm in completely different ways such as by: (1) not using it at all for the meta-learning components (Rusu et al., 2018), (2) not using learned $\beta$ and $\gamma$ parameters at all while still using estimated population means and variances during inference Zhang et al. (2019), or (3) meta-learning $\beta$ and $\gamma$ while only using batch statistics for the normalization (Finn et al., 2017;

---

[7]The examples in the PASCAL dataset are known to be more challenging than the FSS-1000 dataset (Wei et al., 2019). From visual inspection of the two datasets, it is also clear that the PASCAL dataset contains more label noise than the FSS-1000 dataset. For these reasons, the mean IoU values shown in Figure 3, which contain examples from both datasets, are not directly comparable to the results shown in Table 2, which contain examples only from FSS-1000. Furthermore, as discussed in the appendix in C, we did not tune $\omega$ in these experiments.

Nichol & Schulman, 2018), or (4) meta-learning $\beta$ and $\gamma$ and also using population estimates of the mean and variance, as done conventionally when training deep neural networks in the large data regime, which is the approach that we adopt and find to be most useful. Finally, another interesting question to address would be to evaluate how EfficientLab performs on more standard many-shot multi-class image segmentation problems such as the CityScapes dataset (Cordts et al., 2016).

In conclusion, we have shown in this work that the optimal hyperparameter configuration for the update routine may not be the same configuration used during meta-learning. These findings are supported by our theoretical analyses which show that MAML-type algorithms minimize the empirical risk on the training set of a fixed update routine and the initialization $\theta_0$, but do not natively guarantee that the update routine's hyperparameters are optimal. We suspect that improvements realized by relation networks (Wei et al., 2019; Zhang et al., 2019; Rusu et al., 2018), models that learn to generate parameters conditioned on the training data (Rusu et al., 2018; Shaban et al., 2017), and models with learned learning rates (Li et al., 2017; Antoniou et al., 2018) directly leverage information on *how* to adapt given a few-shot sample of labeled examples. We also suspect that the previous work in Mishra et al. (2018) may have found MAML-type algorithms to overfit when applied to high dimensional parameter spaces due to lack of regularization and lack of an empirical risk minimization of the update routine's hyperparamters. It is our hope that our empirical analyses and formalization of the generalization error of meta-learning systems lead to better explanations of why some meta-learning systems work better than others in different problem spaces. Lastly, we hope that this work draws, what we argue is necessary, attention to the open problem of building learning systems that can unify small and large data regimes by gaining expertise and integrating new information as more data becomes available, much as people do.

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

## A    EXAMPLE PREDICTIONS

We have included here a visualization of a small random sample of predictions on test examples $D_{test}$ from test tasks $S^{test}$ that were never seen during meta-training.

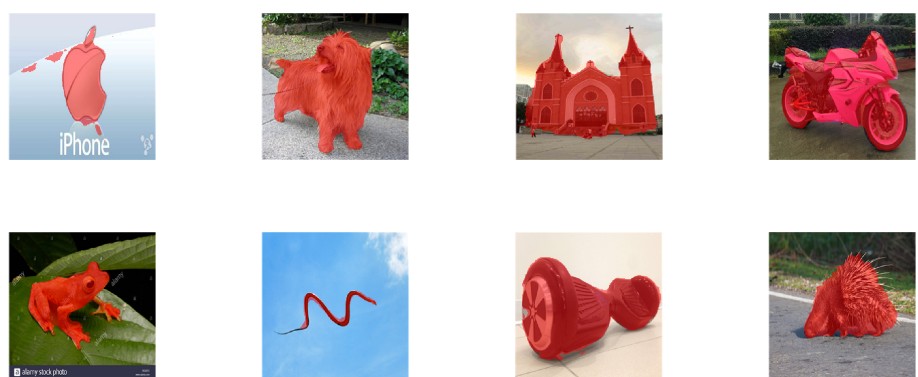

Figure 4: Randomly sampled example 5-shot predictions on the test images from test tasks. Positive class prediction is overlaid in red. From left to right, top to bottom, the classes are `apple_icon`, `australian_terrier`, `church`, `motorbike`, `flying_frog`, `flying_snakes`, `hover_board`, `porcupine`

## B    FP-K DATASET

Table 3 contains the five tasks in PASCAL-5[i] that have direct analogs in FSS-1000. Each row contains the name of a task in FSS-1000 and PASCAL-5[i], respectively. We combine all examples for synonymous tasks. During evaluation, we simply randomly sample 20 test examples, and sample a training set of $k$ examples over the range: [1, 5, 10, 50, 100, 200, 400]. For more details on our training and evaluation procedures see C.

| PASCAL-5[i] Task | FSS-1000 Task |
|---|---|
| aeroplane | airliner |
| bus | bus |
| motorbike | motorbike |
| potted_plant | potted_plant |
| tvmonitor | television |

Table 3: PASCAL-5[i] tasks with FSS-1000 analog.

## C  FP-K EXPERIMENTAL DETAILS

In this section, we describe our testing protocol and simple hyperparameter choices when training the ImageNet and randomly initialized and the FOMAML initialized EfficientLab network. For each tuple of (initialization, k-training shots) we randomly sample 20 examples for a test set, $\mathcal{D}_{test}$ for the task and train on k labeled examples $\mathcal{D}_{tr}$. We repeat this random sampling and training process 4 times for each of the 5 tasks, yielding 20 evaluation samples per (initialization, k-training shots) tuple. For training both networks on the available labeled examples, we simply use the number of iterations [1, 5, 10, 25, 50, 100, 200] for each k in the set of k-shot tasks [1, 5, 10, 50, 100, 200, 400], respectively. We acknowledge that it is likely possible that both networks could achieve better performance if their update routine's hyperparameters are empirically optimized or conditioned on the k examples in $\mathcal{D}_{tr}$.

## D  HYPERPARAMETER TUNING METHODOLOGY

We apply the insight discussed in section 3.2 that the loss of a meta-learning algorithm with a *fixed* initialization $\theta_0$ is a function of the update routine $U$ hyperparameters $\omega$ to develop a simple update hyperparameter optimization (UHO) algorithm. This algorithm is outlined in pseudocode in 1. The routine samples $n$ hyperparameter values within a predefined, broad range, evaluates each of the hyperparameters on a sample of tasks, $\mathcal{T}_{val}$. After evaluating all $\mathcal{T}_i$ in $\mathcal{T}_{val}$, UHO defines a range around the $x\%$ best hyperparameter configurations and samples from that space, repeating until a predefined computational budget is exhausted or the expectation of the loss is no longer reduced. Finally, the best configuration of the hyperparameters that was seen is returned. This algorithm can be a viewed as a variant of Sequential Halving (Jamieson & Talwalkar, 2016).

Because the effects of the learning rate are intertwined with the number of gradient updates, we also leverage early stopping (ES) to decrease runtime and to estimate the optimal number of gradient steps, $\hat{j}^*$, when adapting to a new task. The use of early stopping in this way is purely an implementation optimization that reduces the search space that is explored when tuning the hyperparameters $\omega$. We could have simply randomly sampled the number of iterations, but this would have lead to many wasted tests of combinations of learning rates and gradient steps such as when both the learning rate and the number of gradient updates would be high. We use a patience of 3 steps when using ES, meaning that if the mean IoU on $D^{test}$ did not improve for 3 gradient updates, we stop running SGD and, in our actual code, we also return the number of updates with the best performance on $D^{test}$ on line 6 of the UHO pseudocode.

## E  FSS-1000 TEST TASKS

The set of test tasks that the meta-learning systems are evaluated on in this paper are available in the FSS-1000 github repository https://github.com/HKUSTCV/FSS-1000.

**Algorithm 1** Update hyperparameter optimization (UHO) via random search and successive decrease in search space. Returns the estimated optimal configuration of hyperparameters $\hat{\omega}^*$, such as the learning rate $\hat{\alpha}^*$ and number of gradient steps $\hat{j}^*$, that minimize the empirical loss $p(\mathcal{T})$

**Require:** $\theta_0$: an initialization
**Require:** $f$: a model parameterized by $\theta$
**Require:** $p(\mathcal{T})$: a distribution over tasks
**Require:** $U$: an update routine with hyperparameters $\omega$
**Require:** $\mathbb{S}$: a set of ranges for hyperparameters in $\omega$ to search through
**Require:** $x$: a floating point value in $(0, 1)$ to decrease the search ranges by
**Require:** $b$: the number of iterations to refine the search space, which defines a computational budget
**Require:** $n$: the number of configurations to sample per refinement
 1: Initialize list of metrics, `metrics`
 2: Initialize list of sampled hyperparameter configurations, `parameters`
 3: **for** $i = 0$ **to** $b$ **do**
 4:     **for** $j = 0$ **to** $n$ **do**
 5:         $\omega_s \leftarrow$ `sample`$(\mathbb{S})$
 6:         `metric`, $\hat{j}^* \leftarrow$ `evaluate`$(U, \omega_s, f, \theta_0, p(\mathcal{T}))$   $\triangleright$ Evaluatethe expectation of the loss over $p$, returning loss and best seen number of gradient steps from ES.
 7:         `metrics.append(metric)`
 8:         $\omega_s \leftarrow \omega_s + \hat{j}^*$                      $\triangleright$ Add number of steps returned by ES to the set $\omega_s$
 9:         `parameters.append`$(\omega_s)$
10:     **end for**
11:     $\mathbb{S} \leftarrow$ `best_x_configurations`$(x, \texttt{metrics}, \texttt{parameters})$       $\triangleright$ Get the x% of the best hyperparameters configurations identified by metrics.
12: **end for**
13: **return** $\underset{\omega}{argmax}(\texttt{metrics})$

