# OpenReview forum: "Meta-Learning Initializations for Image Segmentation"
_ICLR.cc/2020/Conference — Reject_

### Official Review · AnonReviewer3 · 2019-10-23
**Official Blind Review #3**

**Rating:** 3

**Review:**

The paper examines the performance of MAML, FOMAML, and Reptile gradient based meta-learning algorithms on the task of semantic image segmentation. The paper proposes to do black box optimization (successive halving) on the hyperparameters of the inner loop of the gradient meta-learners for improved performance. The paper proposes some modification on the segmentation model architecture (no ablation study presented). Finally, it is shown that pre-training using meta-learning on similar segmentation tasks works better then just using ImageNet based model pre-training.

In its current form I suggest to reject the paper and urge the authors to improve it according to the following points:

1. In parts (specifically the intro and some other earlier parts) the paper is very well written, but the later parts, the description of the model modifications (did you consider to add an architecture diagram?), the details of the experiments, the punch-line of the theory development that has been attempted, etc are not very clear and hard to follow. I suggest the authors to improve the readability of these parts, add some helpful / motivating diagrams and examples (perhaps some qualitative results too?), state more clearly what is used for meta-training? how it is made sure that meta-testing is done on a separate set of categories? (I did not see this split in the Appendix) and etc

2. My main concern is novelty. As it stands, the current novelty proposition is: black-box optimization of LR and number of iterations in MAML style meta learning (hardly novel), architecture modifications (no ablation study if these help or not), small improvement on FSS-1000 5-shot test (what about other shots? still not sure about the splits), and showing meta-training on similar tasks is better then not doing it (that is using ImageNet pre-trained backbone for init) - again hardly a novel insight. For the last point, saying that meta-learned model was initialized from scratch does not cut it, as it was meta-trained on massive data that is more related to the test tasks then the ImageNet.

I suggest the authors to mainly focus on 2, although making the writing clearer and better is also very important for a high quality paper.

**Experience Assessment:**

I have published one or two papers in this area.

**Review Assessment: Checking Correctness Of Derivations And Theory:**

I assessed the sensibility of the derivations and theory.

**Review Assessment: Checking Correctness Of Experiments:**

I assessed the sensibility of the experiments.

**Review Assessment: Thoroughness In Paper Reading:**

I read the paper thoroughly.

---

> ### Author Response · Authors · 2019-11-15
> **Thank you for your excellent feedback**
>
> Firstly, thank you for taking the time to review our paper and provide helpful feedback. We have incorporated your feedback and the revised manuscript is much stronger now.
>
> Regarding your concerns, thank you for the stating your concerns clearly around novelty. Our novelty proposition is stated in our general response to reviewers above
>
> 1 and 5 shots are the canonical number of training examples per class in the few shot literature. So for easy comparison, we have included results on the FSS-1000 dataset for those. Furthermore, each class in FSS-1000 only has 10 labeled images total so the absolute maximum you could evaluate via cross-validation would be 9 training shots. One of our central claims is that understanding how meta-learning systems scale to many-shot regimes deserves more attention, so we have released the small FP-k benchmark. We believe that this is also a novel contribution.
>
> Thank you again for your excellent feedback and we look forward to hearing from you!

---

> ### Author Response · Authors · 2019-11-15
> **Regarding visualizations**
>
> Thank you for noting that visualizations of the EfficientLab architecture and qualitative predictions would help. We have added both. The EfficientLab diagram is included in section 4 and example predictions are included in appendix section A.

---

### Official Review · AnonReviewer1 · 2019-10-24
**Official Blind Review #1**

**Rating:** 3

**Review:**

Summary - The paper first makes the observation that training algorithms and architectures for meta-learning have become increasingly specific to the few-shot set of tasks. Following this, the authors first investigate if it’s possible to learn good initializations for dense structured prediction tasks such as segmentation (across arbitrary amounts of available input data). Concretely, the claimed contributions of the paper include -- (1) extension and analysis of first order MAML like approaches to image segmentation; (2) using a formalized notion of the generalization error of episodic meta-learning approaches to decrease error on unseen tasks; (3) doing this via a novel neural network parametrically efficient segmentation architecture and (4) empirically comparing meta-learned initializations with ImageNet pre-trained initializations with increasing training set sizes.

Strengths

- Apart from the flaws mentioned under weaknesses, the paper is generally easy to follow. While it’s somewhat hard to understand the motivations and the concrete contributions made by the paper, sections are more-or-less well-written.
Using the proposed hyper-parameter search scheme over first-order MAML approaches demonstrates improvements over not baselines which do not use the same and baselines which do not involve meta-learning.

Weaknesses

The paper has some major weaknesses that affect the clarity of the points being conveyed in several sections. These weaknesses form the basis of my rating and addressing these would not only help in adjusting the same but would also help in improving the current version of the paper significantly. Highlighting these below:

- The paper claims to make several contributions but it’s hard to concretely understand them in several sections. For instance, the abstract mentions -- ‘’A natural question that ….. human level performance in both.” The statement is slightly unclear to me -- is the intended sentiment the fact that the goal should be to develop a single algorithm that works well for both few-shot and many-shot settings? If so, why should that be the case? Essentially, what is the limiting factor being identified that restricts few-shot approaches from performing well in many-shot settings? Maybe the statement could be framed better but in it’s current form it’s unclear what is being conveyed. When this is mentioned again in the introductory section, it is followed by a statement indicating that meta-learning an initialization is one solution. Why is this surprising? Maybe I’m mis-understanding the motivation behind the claim. Could the authors clarify this?

- Similarly, it’s unclear what question (3) in the introduction is trying to address. Which “data” (training / testing set of tasks) is the fixed update policy not conditioned on? Could the authors clarify this?

- The description of the single update hyper-parameter optimization (UHO) is hard to understand in Sec. 4. -- specifically the text surrounding eqns (5) and (6). The transition from Eqn (5) -> Eqn (6) is unclear. Could the authors clarify this clearly? This section is further referred to in subsequent sections as a supporting basis for some of the obtained results (specifically, the last para on page 6)

Reasons for rating

I found certain sections of the paper particularly hard to understand and interpret. I would encourage the authors to address these more clearly in the responses. The highlighted strengths and weaknesses of my rating and addressing those clearly would help in improving my current rating of the paper.


**Experience Assessment:**

I have read many papers in this area.

**Review Assessment: Checking Correctness Of Derivations And Theory:**

I assessed the sensibility of the derivations and theory.

**Review Assessment: Checking Correctness Of Experiments:**

I assessed the sensibility of the experiments.

**Review Assessment: Thoroughness In Paper Reading:**

I read the paper at least twice and used my best judgement in assessing the paper.

---

> ### Author Response · Authors · 2019-11-15
> **Thank you for your excellent feedback**
>
> Firstly, thank you again for taking the time to review our paper and provide helpful feedback. We will address your feedback point by point.
>
> In regards to: " Similarly, it’s unclear what question (3) in the introduction is trying to address. Which “data” (training / testing set of tasks) is the fixed update policy not conditioned on? Could the authors clarify this?"
> Thank you for pointing out the ambiguity here. The hyperparameters of the update routine of MAML-type algorithms are not automatically a function of any data, but are hardcoded by researchers and guessed via trial and error on validation datasets. But, what we specifically are concerned with is that the update routine’s hyperparameters are not conditioned on the few shot examples for the task, $\mathcal{T}_i$ being adapted to. We utilize random search with successive halving of the search space to estimate the update routine hyerparamaters that maximize the expectation of the intersection over union. We have revised research question 3 to be more explicit: Are MAML-type algorithms hindered by having a fixed update policy for training and testing tasks that is not conditioned on the available labeled examples for a new task?
> We also discuss this clearer and more formally in a new section: 5.3 HYPERPARAMETER SELECTION

---

> > ### Author Response · Authors · 2019-11-15
> > **Regarding motivation for scaling few-shot learning systems to many-shot domains**
> >
> > Regarding your concerns: "The paper claims to make several contributions but it’s hard to concretely understand them in several sections. For instance, the abstract mentions -- 'A natural question that ….. human level performance in both.' The statement is slightly unclear to me -- is the intended sentiment the fact that the goal should be to develop a single algorithm that works well for both few-shot and many-shot settings?
> > Yes, that is exactly the goal.
> > "If so, why should that be the case?"
> > Thank you for noting the justification is not self-evident. We have added this sentence to the introduction: “This property is desirable because training and maintaining multiple models is more cumbersome than training and maintaining one model.”

---

> > ### Author Response · Authors · 2019-11-15
> > **Regarding clarity**
> >
> > Thank you for pointing out that the narrative that ties the paper together is unclear. In our revised version of the manuscript, we have made sure the research questions, results, and primary contributions are much more cohesive.

---

### Official Review · AnonReviewer2 · 2019-10-24
**Official Blind Review #2**

**Rating:** 3

**Review:**

This paper proposes to apply MAML-style meta-learning to few-shot semantic segmentation in images. It argues that this type of algorithm may be more computationally-efficient than existing methods and may offer better performance with a higher number of examples. They further propose to perform hyper-parameter search to choose a new learning rate for the inner learning process after optimizing for the network parameters.

As far as I know, this is the first paper to apply gradient-based (i.e. MAML-style) meta-learning to this specific problem. Existing approaches to few-shot semantic segmentation have mostly used multi-branch conv-networks to condition the output on the training examples. This paper shows that (FO)MAML achieves similar accuracy to the FSS-1000 baseline. This is an empirical contribution in itself. The paper also demonstrates that the EfficientNet architecture can be applied to segmentation.

Major concerns:

(1.1) The improvement obtained by the hyper-parameter optimization seems quite marginal (79.0 - 81.4 and 73.3 - 73.9) and there is no study of the variance of the results. The fact that better performance is obtained by tuning the learning rate on the *training* set suggests to me that the improvement might not be significant. You could repeat the experiment by sampling multiple different training and testing sets (with different classes) to estimate (some of) the variance.

(1.2) The formalization in Section 4 is mathematically appealing but seems unnecessary. In the end, the paper is essentially arguing that it's better to use different learning rates (for the inner loop) during meta-training and meta-testing. This seems obvious, since this includes equal learning rates as a special case. The paper proposes to optimize the latter learning rate using a gradient-free method. This argument can be made without considering generalization bounds.

(1.3) One of the central claims of the paper is that "meta-learned representations smoothly transition as more data becomes available." It's not entirely clear what this means. I suppose it means that, with few shots, it should perform as well as existing few-shot methods, but with many shots, it should perform as well as a standard learning algorithm. The paper failed to present any evidence that existing algorithms for few-shot segmentation do not satisfy this property. It would strengthen the argument to include an existing few-shot segmentation algorithm in Figure 2.

(1.4) The details of the experiment in Figure 2 are not clear. Was a different number of iterations used when there are hundreds of examples? Were different hyper-parameters used when optimizing from a pre-trained checkpoint? It would be unfair to use the same hyper-parameters which had been optimized specifically for the meta-learned initialization.

Other issues:

(2.1) It's not clear what it means to achieve human-level performance in the few-shot task and the many-shot task. What is human-level performance at few-shot segmentation? It seems to me that humans are capable of segmenting novel objects (i.e. zero-shot) with almost perfect accuracy. Does this mean that your method should achieve the same accuracy with few- and many-shots?

(2.2) The use of early stopping was unclear. Do you use a fixed number of SGD iterations during training and a variable number of iterations (determined by a stopping criterion) during testing? However, this seems to be contradicted by the statement that the UHO algorithm determined an optimal number of iterations (8) for testing? On the other hand, this seems like too few iterations with hundreds of shots. Maybe the automatic stopping criterion was only used with many shots? Or maybe training proceeds until either the stopping criterion is satisfied or the maximum number of iterations is reached? Furthermore, the early stopping criterion was not specified.

(2.3) Missing reference: Meta-SGD (arxiv 2017) considers a different learning rate for every parameter and updates the learning rates during meta-training .

(2.4) There was no discussion of the running time of different methods. This would be particularly interesting in the many-shot regime. How slow are the RelationNet approaches?

(2.5) It is claimed that Figure 1 demonstrates that  "the estimated optimal hyperparameters for the update routine ... are not the same as those specified a priori during meta-training". However, it seems that the optimal learning rate is awfully close to the dotted blue line (within the variance of the results).

(2.6) For the IOU loss (equation 10), what are the predicted y values? Are they arbitrary real numbers? Do you use a sigmoid to constrain them to real numbers in [0, 1]?

Minor:

(3.1) It is not worth stating the optimal learning rate to more than 3 or 4 significant figures.
(3.2) Use 1 \times 1 instead of 1 x 1.
(3.3) Is there a reference for the Dice score? Where does the name come from?

**Experience Assessment:**

I have published one or two papers in this area.

**Review Assessment: Checking Correctness Of Derivations And Theory:**

I assessed the sensibility of the derivations and theory.

**Review Assessment: Checking Correctness Of Experiments:**

I carefully checked the experiments.

**Review Assessment: Thoroughness In Paper Reading:**

I read the paper at least twice and used my best judgement in assessing the paper.

---

> ### Author Response · Authors · 2019-11-15
> **Thank you for the critical feedback.**
>
> Firstly, thank you for taking the time to review our paper and provide this helpful feedback. We have responded to general themes the reviewers had in the above comment but will also drill into some details here.
>
> Going by your numbered feedback:
> (1.1)
> Thank you for noting the importance of the variance in determining significance. We have rerun the meta-learning experiments with the same test set of tasks from the FSS-1000 authors (this test set hadn’t been released previously but we emailed the authors and they sent us the test set and then released it on their github page) and have reported mean IoU results with 95% confidence intervals. These results are computed by training and evaluating on each of the 240 held out tasks with 2 different random splits of the 10 examples, yielding 480 results data points per meta-learning method and architecture we experimented with.
> (1.2)
> Thank you for critically considering this part of the paper. We agree that it should be obvious that the meta-test results of gradient-based meta-learning depend on the update routine’s learning rate (and other hyperparameters such as number of iterations), but the update routine’s hyperparameters have not been systematically addressed in previous work (Finn et al. 2017, Rusu et al. 2018, Nichol and Schulman 2018) but rather guessed by the researchers. This is concerning because it does not allow us to know what the upper limits of gradient-based meta-learning are. While we agree that it is trivial to show directly that the test-set loss is dependent on the update routine’s hyperparameters, we believe that it is a contribution in its own right to mathematically define the generalization error. As far as we can tell from the papers we have cited, the generalization error has not been stated clearly in terms of the non-computable expected loss over the task-generating distribution. We have made this section a subsection of preliminaries and revised it to be more compact and to clearly and concisely show that generalizing to unseen tasks is a function of both the initialization and the update routine’s hyperparameters.
>
> (1.3)
> Thank you for pointing out these details that our paper lacks clarity around. This is really helpful and echoed by reviewer 1. We did mean that with few shots, it is a desirable property for  a meta-learning approach to perform as well as existing few-shot methods, but with many shots, it should perform as well as a standard learning algorithm. Performing well here would mean having both high accuracy and comparable inference runtime. We have added additional clarity to the paper and removed ambiguous wording. In particular, “smoothly” was a rather ambiguous term, we have revised our submission here to state clearly that we find that our meta-learned initializations continue to provide value as more data becomes available. Regarding providing evidence that other few-shot approaches do not meet this criteria, we did describe general evidence in the Related Work section by mentioning that existing few shot algorithms use model ensembling, relation networks which “typically are $O(n^2)$ in memory consumption and[/or] runtime [depending on the implementation] for the update routine when adapting to the new task, where n=number of training examples”, or iterative optimization modules which require multiple passes through the network per evaluation. We have introduced the FP-k benchmark specifically so that future meta-learnign work can compare the scalability of their approach.
>
> (1.4)
> This is a good point. We did not provide enough details on Figure 2. We are in complete agreement that is not fair to say the same hyperparameters will yield equally maximal results for meta-learned and imagenet+random initializations. We have provided more details on our hyperparameters in the experiments section. Importantly, we have also reframed the discussion of this experiment as a small benchmark that we hope will attract additional inquiry into the empirical performance of learning algorithms that must scale from few to many-shot settings.
>
>
> References:
>
> Chelsea Finn, Pieter Abbeel, and Sergey Levine. Model-agnostic meta-learning for fast adaptation of deep networks. arXiv preprint arXiv:1703.03400, 2017.
> Andrei A Rusu, Dushyant Rao, Jakub Sygnowski, Oriol Vinyals, Razvan Pascanu, Simon Osin- dero, and Raia Hadsell. Meta-learning with latent embedding optimization. arXiv preprint arXiv:1807.05960, 2018.
> Alex Nichol and John Schulman. Reptile: a scalable metalearning algorithm. arXiv preprint arXiv:1803.02999, 2018.

---

> > ### Author Response · Authors · 2019-11-15
> > **continuing previous thread**
> >
> > (2.1)
> > Thanks for pointing this out. We agree that “human-level” is an ill-defined term that is both open to subjective interpretation and complex to define precisely. In the few shot domain, accurate segmentation of an object in unseen images requires skills in both recognizing the object and identifying the pixels that belong to that object. How these skills may be implemented in the human perceptual system is out of the scope of this paper. We have removed all references to “human level” segmentation in the paper, noting human performance only in the first sentence with the general statement that humans have a remarkable capability to learn from a small number of labeled examples and gain expertise as more data becomes available.
> >
> > (2.2)
> > We apologize for the confusion around our use of early stopping. Early stopping is only used inside the UHO algorithm and is entirely an implementation optimization to reduce the runtime of the random search. In our specific application of UHO, it returns the best combination of learning rate and number of iterations seen on the provided tasks. We have provided more details on UHO algorithm and the k-shot learning curves in the appendix and moved the only mention of early stopping to the UHO section in the appendix. We do not use tuned hyperparameters for the k-shot learning curves experiments.
> >
> > (2.3)
> > Thank you for noting that important paper. We were aware of it and that it is cited by many of the works we reference. We didn’t see the direct need to cite it since it is referenced by successor papers that we do cite. Furthermore, we did not experiment with the Meta-SGD methods due to concern regarding an effective doubling in the number of parameters and computational overhead and the lack of a first order approximation. That said, it is important and related work so we have added it to the related work section.
> >
> > (2.4)
> > Runtime is important but was not the focus of this paper. Given that it wasn't the focus we did not have time compute empirical runtimes for both another work and our model. We leave this for future work. Depending on the implementation, RelationNet's can have $O(n^2)$ time costs in the number of training examples.
> >
> > (2.5)
> > This figure is doing double duty. One point we wanted to show is that the representations are not robust to large changes in the update routine’s hyperparameters. Thank you again for noting the importance of showing the variance of results. We have since added these for the meta-test set. The lower bound of the 95% confidence interval of mean IoU is greater than the mean IoU from using the meta-training learning rate, so the improvements do appear to be significant.
> >
> > (2.6)
> > It was noted in the Neural Network Architecture section (which has been renamed to EFFICIENTLAB ARCHITECTURE FOR IMAGE SEGMENTATION) that we use a softmax to constrain the predictions to real numbers in [0, 1].
> >
> > (3.1)
> > Thanks for pointing this out. We’ve corrected the latex.
> >
> > (3.2)
> > Thanks for pointing this out. We’ve corrected the latex.
> >
> > (3.3)
> > We’ve added the reference for the Dice score, which is also called the Sørensen–Dice coefficient, or the F1 score in many modern domains.
> >
> > Thank you again for your excellent feedback and we look forward to hearing from you!

---

### Author Response · Authors · 2019-11-14
**Your feedback has been extremely helpful and we have addressed your points in the revised manuscript and below comment**

Dear Reviewers,

First we want to sincerely thank all the reviewers for their critical responses and taking the time to analyze our work in depth. Your feedback is greatly appreciated and has helped us clarify many components and make the paper more cohesive and all-around stronger. Second, we apologize for not responding sooner; you raised many good points that we wanted to address as best as we could in the available time.

 We have revised the paper throughout such that all concerns that we could attend to in time have been addressed. We have worked to make the primary contributions and the narrative clearer and removed distracting details that were not related to one of our primary contributions. Notably, you will notice that the generalization error section is only a subsection of Preliminaries and is much more compact now. In exchange, we have put additional explanation and experimental ablation results into the proposed architecture EfficientLab.

In response to concerns of the significance of this work, we have rerun all experiments we had time to (including new ablation experiments) with the exact same test set of tasks used by the FSS-1000 authors and have included 95% confidence intervals on our results. We have also provided additional details on the k-shot learning curves and experiments.

We also want to thank you for your concerns regarding novelty and hope that the revised version of the manuscript has a clearer and stronger novelty proposition. We do politely disagree that applying MAML-style algorithms to larger networks and image segmentation is not novel. This point is clearly evidenced by 2 papers from ICLR 2018 stating that MAML-type algorithms are “unproven for the high dimensionality and skewed distributions of segmentation” (Rakelly et al., 2018) and even more clearly by Mishra et al. (2018) who found that MAML “overfit significantly” when using a deeper neural network. Our novelty proposition in relation to MAML-type algorithms specifically is that they do in fact extend to higher dimensional parameters spaces and the skewed distributions of image segmentation when the hyperparameters of the update routine are tuned to minimize empirical error given a fixed meta-learned initialization, $\theta_0$.  In hindsight, this point may seem obvious, but looking at the existing published literature, it is clear that this point has not been investigated in detail. By defining the generalization error of a meta-learning system (which we consider another novel contribution), it is self-evident that the generalization error of meta-learning systems is also a function of how they update based on new data. We also have shown that the EfficientLab neural network architecture is an interesting and useful contribution. In our revised version of the manuscript, we have included ablations around the design choices of the architecture. Lastly, the small FP-k benchmark is a novel contribution which we hope will garner more interest to the study of how meta-learning systems scale as more data becomes available. We will included a table next to figure 3 for clarity as a benchmark tomorrow.

Notation and terminology changes:
We have changed notation in a few places to make the paper clearer, easier to read, and more consistent with other key papers in meta-learning. Specifically we have changed:
 - $\tau$ to $\mathcal{T}_i$ the random variable denoting the generating distribution of a task (such as the generating distribution that would generate image-label pairs for the task black bear).
 - $\mathbb{H}$ to $\omega$ indicating the hyperparameters that are built into the update routine $U$. We do this because using a set indicator was visually a bit more distracting than the variable $\omega$
 - minimize to argmin
 - $\tau_s$ to $\mathcal{D}_s$ denoting an empirical sample of image-mask pairs from a task $\mathcal{T}_i$
 - Lightweight skip decoder to residual skip decoder (RSD) emphasizing that one of the most meaningful changes in the atrous spatial pyramid pooling block is the residual connection

Additional Gaps that have been filled:
We neglected to exclude a few details and citations which we have added to the revision:
 - L2 term to the loss
 - Simple augmentations of training examples during training
 - Citation of loss function which we adapted

We will be responding to your individual points in more depth tomorrow and we hope you will have the time to look at the revised version of the paper. Thank you again for your feedback and time.

Sincerely,
The authors

References:
Kate Rakelly, Evan Shelhamer, Trevor Darrell, Alyosha Efros, and Sergey Levine. Conditional networks for few-shot semantic segmentation, 2018. URL https://openreview.net/ forum?id=SkMjFKJwG.
Nikhil Mishra, Mostafa Rohaninejad, Xi Chen, and Pieter Abbeel. A simple neural attentive meta-learner. In International Conference on Learning Representations, 2018. URL https: //openreview.net/forum?id=B1DmUzWAW.

---

### Decision · Program_Chairs · 2019-12-19

**Decision:**

Reject

**Comment:**

The reviewers reached a consensus that the paper was not ready to be accepted in its current form. The main concerns were in regard to clarity, relatively limited novelty, and a relatively unsatisfying experimental evaluation. Although some of the clarity concerns were addressed during the response period, the other issues still remained, and the reviewers generally agreed that the paper should be rejected.